# Development and psychometric assessment of cutaneous leishmaniasis prevention behaviors questionnaire in adolescent female students: Application of integration of cultural model and extended parallel process model

**Masoumeh Alidosti**[1], **Hossein Shahnazi**[2‡], **Zahra Heidari**[3‡], **Fereshteh Zamani-Alavijeh**[2]*

**1** Student Research Committee, School of Health, Isfahan University of Medical Sciences, Isfahan, Iran,
**2** Department of Health Education and Promotion, School of Health, Isfahan University of Medical Sciences, Isfahan, Iran, **3** Department of Biostatistics and Epidemiology, School of Health, Isfahan University of Medical Sciences, Isfahan, Iran

☯ These authors contributed equally to this work.
‡ HS and ZH also contributed equally to this work.
* fe.zamani@hlth.mui.ac.ir, fe.zamani@gmail.com

**Data Availability Statement:** All relevant data are within the article and its Supporting information files.

## Abstract

### Background

Cutaneous Leishmaniasis (CL) is an important public health issue in at least 83 countries, including Iran. Individuals' behavior modification is believed to be one of the best ways for CL prevention. However, no comprehensive questionnaires have been psychoanalyzed for identification of CL prevention behaviors and its numerous associated factors, as well as the impact of educational messages. Thus, the present study was conducted to develop and psychometrically assess CL prevention behaviors questionnaire in female students.

### Methods

The present study was performed from October 2020 to May 2021 by developing a preliminary questionnaire based on integration of Cultural Model and Extended Parallel Process Model. The questionnaire was completed online by 460 adolescent female students living in endemic areas of Isfahan, Iran. Exploratory factor analysis was performed using SPSS 24 to ensure the construct validity. Internal reliability was assessed via Cronbach's alpha and external reliability was determined using the test-retest method based on the intraclass correlation coefficient (ICC) index.

### Results

The first version of the questionnaire contains 110 items, out of which 82 remained according to content validity ratio (CVR) and content validity index (CVI) criteria. Afterwards, 11 items were removed due to low factor load in the construct validity process using the factor

**Funding:** The authors received no specific funding for this work.

**Competing interests:** The authors have declared that no competing interests exist.

analysis technique. Ultimately, a 71-items questionnaire was developed and 12 factors were extracted from it. According to Cronbach's alpha index, the internal reliability for the questionnaire was 0.877 and the ICC index calculated the external reliability as 0.833.

## Conclusions

Integration of a Cultural Model with individual model was used for the first time to measure the factors related to CL prevention behaviors in this questionnaire; owing to the strength of the factor structure and appropriate psychometric properties, the questionnaire is applicable in the evaluation process of educational interventions concerning CL prevention, especially in female students.

## 1. Introduction

Cutaneous Leishmaniasis (CL), as the most prevalent type of leishmaniasis, is an important public health issue in at least 83 countries, including Iran [1]. The cause of this disease is a protozoan parasite transmitted by sandfly bites [2]. The treatment course of CL is long, complex, and expensive, with certain complications, including permanent skin lesions, developing secondary infections, and psychological and social problems [3, 4]. The annual number of cases suffering from this issue has been estimated to be between 600 000 to 1 million worldwide [4]. With an annual incidence rate of 30,000 cases [5], Iran is one of the seven countries with the highest prevalence of CL [6]. Isfahan, located in the center of Iran on green plains of Zayandeh Rud River, is one of the CL endemic provinces in Iran, which has faced a significant increase in the prevalence of the disease over the last decade. It is particularly observed in the north and northeast of Isfahan province, including wet or rural type (Zoonotic Leishmaniasis or ZCL). ZCL is caused by leishmania major and transmitted by Phlebotomus papatasi [7, 8]. Due to the lack of vaccines to prevent CL, the study of the situation and promoting preventive behaviors in endemic areas have been suggested as one of the most important strategies to control leishmaniasis [9].

### 1.1. The necessity to study CL prevention behaviors in female students

The outcomes of the appearance of scar production on visible parts of the body are more important in women than that those in men [10]. Unsightly wounds and scars on the face of young women are not only psychologically and socially offending for them [11], but also cause stigma and, in particular, reduce their chances of getting a partner for marriage [3, 12]; on a number of occasions, this stigma leads to their isolation [10]. As relevant research literature indicates, adolescent girls suffer more compared to others in terms of psychological adverse effects and experience of stigma caused by CL [10, 11]. On the other hand, most adolescent girls are students. Therefore, paying attention to CL prevention behaviors in this group can be a priority. In this regard, a reliable and valid tool is needed for the assessment and evaluation of educational programs related to preventive behaviors.

### 1.2. Theoretical framework selection

With the purpose of identifying CL prevention behaviors, some researchers have used the Health Belief Model (HBM) and individual predisposing factors, such as perceived susceptibility, perceived severity, and self-efficacy, to study some parameters, such as attitude and

intention used the BASNEF model [13, 14]. Each of these frameworks includes only a limited number of factors and variables; they do not consider the impact of different types of educational messages. However, in addition to the possibility of measuring the above-mentioned variables, the Extended Parallel Process Model (EPPM) provides the possibility to measure response efficacy and impact of different types of educational messages on the form of audience reactions, including "danger control" and "fear control" [15]. According to the EPPM, it is more likely that individuals efficiently apply health-associated behaviors for danger control providing that they believe their health is at serious risk and trust the efficacy of coping strategies and their self-efficacy to adapt those strategies and preventative behaviors; under this condition, there would not be fear control process stages [15–17]. Although EPPM appears as an appropriate model for the study of individual factors of behavior, the role of enabler factors and social environment should also be considered in identifying CL prevention behaviors [18]. According to studies performed in the form of PRECEDE and PEN-3 models, all the three factors of perceptions, enablers, and nurtures have an important role in the continuation of behaviors, including CL prevention behaviors [5, 19]. The PEN-3 Model consists of three domains: (1) cultural identity, (2) relationships and expectations, and (3) cultural empowerment. Each domain includes three factors that form the acronym PEN, namely the following: person, extended family, neighborhood (cultural identity domain); perceptions, enablers, and nurturers (relationship and expectation domain); positive, existential, and negative (cultural empowerment domain) [19]. This model initially identifies positive, neutral, and negative entities for observing behavior in society, which could be conducive to the prevention of indigenous diseases [19], In addition to perceptions (perceived susceptibility, perceived severity, and self-efficacy,..), this cultural model considers incentives or "nurtures" and also "enablers", for preventive behaviors [20]. According to these explanations, developing a questionnaire based on the integration of PEN-3 Model and EPPM (Fig 1) would be useful for examination of a wider range of social and individual factors relevant to CL prevention behaviors and evaluation of the impact of interventions.

Therefore, based on the above points, the present study aimed to develop and psychometrically assess Cutaneous Leishmaniasis prevention behaviors questionnaire with application of integration of cultural PEN-3 Model and Extended Parallel Process Model in adolescent female students in endemic areas.

## 2. Materials and methods

The present methodological research was conducted from October 2020 to May 2021 (code of ethics: IR.MUI. RESEARCH. REC.1399.430). the target population for instrument design and psychometric were female students aged 12 to 17 years in the endemic areas of Isfahan, Iran. The participants signed informed consent form prior to the beginning of the study. Fig 2 shows the stages of development and psychometric evaluation of the Cutaneous Leishmaniasis prevention behaviors questionnaire (Fig 2).

### 2.1. Initial questionnaire item generation

The item generation was performed via a survey in the social and cultural conditions of the community and by reviewing the results CL studies around the world [21] in addition to the related questionnaires [9, 14, 18, 22]. Afterwards, the pool of questions with 110 items was made in Persian Version, which is the native language of the target group. The itemization and dimension determination of the questionnaire was performed based on the integrated PEN-3 Model and EPPM according to the guidelines and scientific texts related to these models [16, 19, 20, 23, 24].

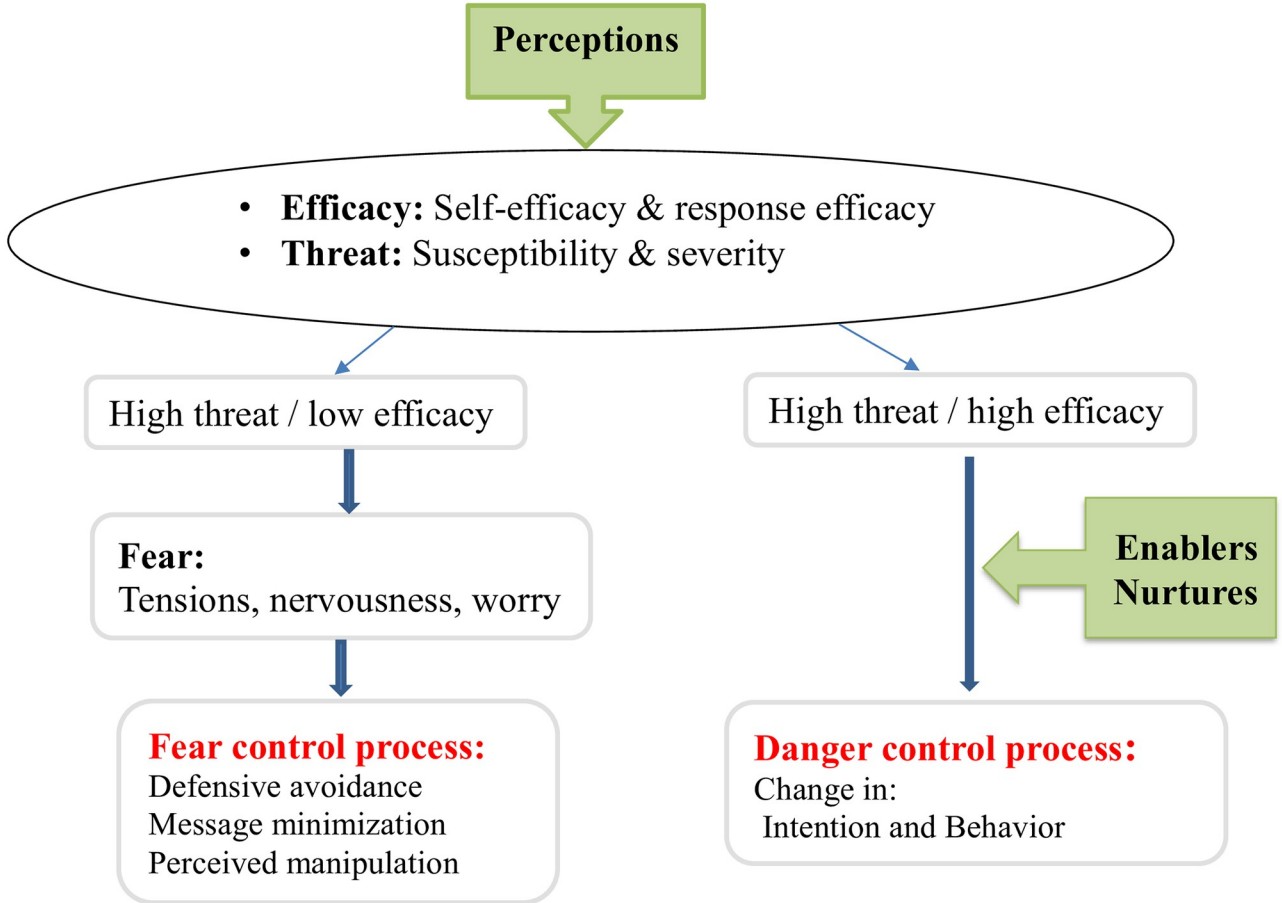

**Fig 1. The pattern of PEN-3 and EPPM models integration used in the present paper.**

## 2.2 Content validity assessment

The content validity was evaluated qualitatively by eight experts (five health education specialists and three healthcare workers) after a careful review of the items with an emphasis on grammar, wording, correct item placement, and scoring. Their written corrective views were applied in the questionnaire.

For the content validity which was performed quantitatively, 14 experts (10 health education specialists and four healthcare workers) were asked to choose "Essential", "Useful but not essential", and "No Necessary" for each item based on the following formula: CVR = [ne − (N/2)] / (N/2). Following the content validity ratio (CVR) calculation, the items with a CVR of higher than 0.51 were accepted according to the Lawshe formula [25]. The same 14 experts were then asked to review and specify the following features for each item in a four-point Likert scale: 1. relevance (1 = irrelevant, 2 = relatively relevant, 3 = relevant, 4 = quite relevant), 2. simplicity (1 = not simple, 2 = relatively simple, 3 = simple, 4 = quite simple), and 3. clarity (1 = not clear, 2 = relatively clear, 3 = clear, 4 = quite clear).

Subsequently, content validity index (CVI) was calculated based on the following formula: CVI = the number of experts who scored the items 3 or 4 / the total number of the experts. The CVI value above 0.79 was considered acceptable for the items [25]. The content validity

**Initial questionnaire item generation**: by survey the community social and cultural conditions and Literature review

110 items

**Content validity assessment:** The content validity qualitatively, by eight experts (five health education specialists and three healthcare workers) and the content validity performed quantitatively by10 health education specialists and four healthcare workers.

82 items

**Face validity assessment:** The face validity qualitatively, by 10 students and eight experts (five health education specialists and three healthcare workers) and the face validity performed quantitatively by10 adolescent female students.

82 items

**Construct validity assessment:** The exploratory factor analysis performed in a cross-sectional study by sampling of 460 adolescent female students.

71 items

**Assembled a questionnaire with 71 items, with high reliability and homogeneity.**

**Fig 2. Stages of development and psychometric assessment of cutaneous leishmaniasis prevention behaviors questionnaire.**

control with qualitative and quantitative methods made it possible to eliminate unnecessary and irrelevant questions. The items which were considered ambiguous by the experts were reformulated based on the suggested refinements, and questionnaire entered the next stage with 82 items.

### 2.3. Face validity assessment

The face validity was evaluated qualitatively by the eight above-mentioned experts and the participation of five students via telephone (face-to-face visit was impossible due to the outbreak of COVID-19). After providing the participants with the necessary explanations about the research, their comments on the comprehensibility of each questionnaire item were recorded and the necessary corrections were made. A revised version was then presented to five other students who were asked to express their opinion on the clarity of the items. The results revealed that no more corrections were needed for the items [26].

Thereafter, 10 students were asked to comment on the importance level of each item to quantify the face validity of the questionnaire on a five-point Likert scale from "not important at all" (score = 1) to "very important" (score = 5). Afterwards, the impact score of each item was determined by calculating the impact factor in relative frequency; all the items were

deemed appropriately and kept in the questionnaire since they got a score greater than 1.5 [25].

## 2.4. Construct validity assessment

**2.4.1. Exploratory factor analysis.** The exploratory factor analysis method was employed in a cross-sectional study to identify the main factors of the questionnaire and ensure its validity. At this stage, female students living in endemic areas of Isfahan, Iran, who agreed to participate in the study, were included in the study through an available sampling method. Structural equation models were used herein for data analysis and in these studies, the sample size is usually considered to be five to 10 times of the free model parameters [27]; thus, the sample size per item was selected to be five.

## 2.5. Internal and external reliability assessment

Cronbach's alpha was calculated to determine internal reliability while ICC (intraclass correlation coefficient) was used for the calculation of external reliability.

## 2.6 Data collection for factor analysis

Since the schools were closed due to the COVID-19 outbreak, the questionnaire was applied in https://porsline.ir to be completed by the students. For this purpose, the short link (yun.ir / il8i51) of the questionnaire was made from its original link (https://survey.porsline.ir/s/ ENrbOQE) and it was sent to the high school first-grade female students through virtual networks. In the first part of the questionnaire and after introducing the researcher and other colleagues, some explanations were provided about the purpose of the study, how to participate in the study, and confidentiality of the information; the participants were also informed that they could leave the study whenever they wanted to and that their participation was optional. In the process of answering the questionnaire, they would move throughout the items provided that the consent box was marked.

Two months later, the external reliability of the instrument was calculated via test-retest method. To this end, the same questionnaire was prepared in https://porsline.ir and its link (yun.ir/zzqodb) was sent via SMS to the parents of the students who completed the first series of the questionnaire and voluntarily entered the phone number of one of their parents. They were asked to complete the second questionnaire and the link was deactivated after 31 individuals completed the questionnaire.

## 2.7. Statistical analysis method

SPSS ver.24 was utilized for statistical analysis. For this purpose, the adequacy of the sample size and the correlation between the extracted factors were evaluated with the Kaiser-Meyer-Olkin (KMO) and Bartlett's tests. In addition, interpretable factors were identified using exploratory factor analysis (EFA) with varimax rotation. The factors were then identified by the cut-off point of 1.5 for Eigenvalue and cut-off point factor loads were considered 0.3. The Cronbach's alpha values between 0.70 and 0.95 were satisfactory regarding the external reliability while the internal reliability was determined using intraclass correlation coefficient (ICC) with a two-way random method and a confidence interval of 0.95; a coefficient above 0.70 was considered to have an excellent stability [25].

## 3. Results

A total of 460 female high school students completed the questionnaire, among whom 374 (74.4%) were natives and 88 (17.6%) were non-natives living in the area.

### 3.1. Results of the preliminary

The initial version of the questionnaire had 110 items; the perception section on the Likert scale comprised strongly agree, agree, disagree, and strongly disagree and included 20 questions on perceived susceptibility; there were also nine questions on perceived severity, 10 on self-efficacy, and nine on response efficacy. Six questions of nurture, 13 of enablers, and eight of fear were also on the five-point Likert scale (zero, low, medium, high, very high). Fear control questions were in the four-point Likert ranging from strongly agree to strongly disagree. They included six questions about defensive avoidance, three for message minimization, and five for perceived message manipulation. Danger control questions included five questions concerning attitude on a four-point Likert scale (from strongly agree to strongly disagree), four questions about the intention on a five-point Likert scale (from this month, from next month, after next six months, from next year, never), and 12 questions regarding behavior on a four-point Likert scale (never, sometimes, most of the time, always).

### 3.2. Results of content and face validity assessment

The items with a content validity index (CVI) and content validity ratio (CVR) lower than 0.79 and 0.51, respectively, were removed and ultimately, the content and face validity of the questionnaire was confirmed with 82 items.

### 3.3. Results of construct validity assessment (exploratory factor analysis)

The result of the KMO test showed that the data volume for EFA was desirable. The value of this index was equal to 0.896. Bartlett's test was significant (P <0.001, df: 2485) and suggested a sufficient correlation between the variables. At this stage, 11 items were removed due to low factor load and 71 items were entered into the final analysis. Employing Scree diagram and based on the number of common factors and special values above one, 12 interpretable factors were obtained. The factor load of the items was in the range of 0.422 to 0.889 and the total variance of the 12-factor model was 60.50% (Table 1). The factors were named according to the concepts obtained from the loaded items in each factor and also by reviewing the relevant research literature as follows:

- The first factor, "Perceived efficacy", with the variance of 20.949, consisted of 15 items and expressed the perception of the individuals concerning the effectiveness of the recommended behaviors in preventing CL, as well as their understanding of their ability to perform the aforementioned behaviors.

- The second factor, "Behavior", with the variance of 9.107, consisted of nine items and measured the preventive behaviors of CL.

- The third factor was "Message Minimization and Perceived Manipulation" with the variance of 6.006. It comprised eight items and measured the person's perception of the value and accuracy of CL-related messages. As shown in Table 1, one of the items in the fourth factor (I do not care about the written and visual material about the risks of the seeker) was also included in this group since it is conceptually more similar to this category.

**Table 1. Items of the questionnaire of the CL preventive behaviors and factors related to the integration of PEN-3 and EPPM with mean, Std. Deviation, CVR, and CVI.**

| Component | Row | Item | Mean | Std. Deviation | Cronbach's Alpha if Item Deleted | I-CVI | CVR |
|---|---|---|---|---|---|---|---|
| 1) Perceived Efficacy | 1 | I can ask my family to install a suitable net in door frames, window frames, and air conditioner vents. | 2.43 | .668 | .874 | 1 | 1 |
| | 2 | I can ask my family to provide insect repellents, such as spray and repellent pen. | 2.40 | .732 | .873 | 1 | 1 |
| | 3 | I can ask my family to repair the cracks on the walls of the house. | 2.27 | .781 | .873 | 1 | 1 |
| | 4 | My request to the family for installing a suitable net for doors and windows and air conditioner vents will be effective. | 2.32 | .729 | .873 | 1 | 1 |
| | 5 | I can use an insect repellent pen or ointment properly. | 2.38 | .727 | .874 | 1 | 1 |
| | 6 | I can use a mosquito net to sleep outdoors. | 2.36 | .776 | .874 | 1 | 1 |
| | 7 | I can tell my family to put the garbage out of the house during the collection hour. | 2.40 | .718 | .874 | 1 | 1 |
| | 8 | My family can contact the municipality or relevant organizations to collect construction waste. | 2.23 | .810 | .873 | 1 | .85 |
| | 9 | My request from the family to repair cracks in the walls of the house will be effective. | 2.15 | .772 | .873 | 1 | 1 |
| | 10 | The use of insect repellents, such as ointment and repellent pen, is useful to prevent CL. | 2.37 | .709 | .874 | 1 | 1 |
| | 11 | My request to my family to provide insect repellents, such as a repellent pen, will be effective. | 2.31 | .678 | .874 | 1 | 1 |
| | 12 | I can cover most parts of the body during biting times. | 2.34 | .777 | .874 | 1 | .85 |
| | 13 | The use of mosquito nets at rest is useful to prevent CL. | 2.44 | .672 | .874 | 1 | 1 |
| | 14 | It would be effective for my family to contact the municipality or relevant organizations to collect construction debris. | 2.07 | .857 | .874 | 1 | 1 |
| | 15 | I can avoid sleeping outdoors if I do not have a mosquito net. | 2.20 | .866 | .874 | 1 | 1 |
| 2) Behavior | 1 | I suggest the family to install a suitable net in front of the doors and windows. | 2.15 | .978 | .873 | 1 | 1 |
| | 2 | I ask my family to contact the municipality or the relevant organization to collect construction debris in the neighborhood. | 1.84 | 1.11 | .874 | 1 | 1 |
| | 3 | I ask my family to provide insect repellents, such as pen or insect repellent ointment. | 2.11 | 1.018 | .872 | 1 | 1 |
| | 4 | I ask my family to take the garbage out of the house during the collection hour. | 2.24 | .972 | .872 | 1 | 1 |
| | 5 | I suggest the family to repair the cracks on the walls of the house. | 1.86 | 1.070 | .872 | 1 | 1 |
| | 6 | If I want to get some rest outdoors, I use a mosquito net. | 1.97 | 1.084 | .873 | 1 | 1 |
| | 7 | I use insect repellent when leaving the house when there is a possibility of a bite. | 1.87 | 1.102 | .873 | 1 | 1 |
| | 8 | When leaving home, I wear appropriate clothing with long sleeves when there is a possibility of a bite. | 2.29 | .912 | .875 | 1 | 1 |
| | 9 | I pay attention to the prevention of CL. | 1.92 | .929 | .874 | 1 | 1 |

(*Continued*)

**Table 1.** (Continued)

| Component | Row | Item | Mean | Std. Deviation | Cronbach's Alpha if Item Deleted | I-CVI | CVR |
|---|---|---|---|---|---|---|---|
| **3) Message Minimization and Perceived Manipulation** | 1 | Messages related to CL are misleading. | .82 | .835 | .880 | .92 | .71 |
| | 2 | CL messages have been distorted and manipulated. | .80 | .828 | .881 | .89 | .71 |
| | 3 | The words and pictures of the CL are wrong. | .84 | .818 | .879 | .97 | .85 |
| | 4 | The words and pictures of the CL are false. | .69 | .759 | .879 | .97 | .85 |
| | 5 | CL messages are designed for other purposes. | .87 | .813 | 879 | .82 | .57 |
| | 6 | Messages related to CL are obligatory and authoritarian. | .98 | .877 | .880 | .92 | 1 |
| | 7 | The texts and pictures of the CL are exaggerated. | 1.20 | .889 | .879 | .97 | .85 |
| | 8 | I do not care about the written and visual content about the dangers of the CL. | 1.15 | .905 | .881 | .94 | 1 |
| **4) Defensive Avoidance of the Message** | 1 | I do not want to talk about the CL. | 1.39 | 1.020 | .880 | .97 | 1 |
| | 2 | I do not want to hear about the CL. | 1.37 | 1.021 | .880 | .94 | 1 |
| | 3 | I prefer not to think about the side effects of CL. | 1.48 | 1.009 | .879 | .97 | 1 |
| | 4 | I avoid viewing images related to the CL (such as movies, posters, and photos). | 1.36 | 1.012 | .879 | .97 | 1 |
| | 5 | I do not follow the messages and content on social media about the CL. | 1.49 | .980 | .881 | .97 | 1 |
| **5) Nurture** | 1 | School teachers encourage me to take CL prevention measures. | 1.70 | 1.275 | .874 | 1 | .85 |
| | 2 | The school health instructor encourages me to take preventive measures against CL. | 1.78 | 1.296 | .874 | 1 | .85 |
| | 3 | Health professionals encourage me to take CL preventive measures. | 1.92 | 1.224 | .875 | .94 | .85 |
| | 4 | My friends and peers encourage me to take preventive measures against CL. | 1.76 | 1.284 | .873 | 1 | .85 |
| | 5 | Family members encourage me to take steps to prevent CL. | 2.73 | 1.199 | .873 | 1 | .85 |
| **6) Perceived Severity** | 1 | If I get CL, I have to endure a long and painful treatment. | 2.16 | .803 | .876 | .97 | 1 |
| | 2 | If I get CL, its wounds will remain for the rest of my life. | 1.93 | .956 | .876 | .94 | 1 |
| | 3 | If I get CL, my beauty is in danger. | 2.35 | .780 | .875 | 1 | 1 |
| | 4 | If I get CL, it will cost me and my community a lot. | 1.88 | .913 | .875 | .97 | 1 |
| | 5 | CL can cause unpleasant and ugly scares. | 2.52 | .684 | .875 | .94 | .85 |
| | 6 | If I get CL, its wound may become severely infected. | 2.28 | .766 | .875 | .82 | .85 |
| | 7 | CL wounds prevent my friends from contacting me. | 1.70 | 1.001 | .876 | .94 | .85 |
| **7) Fear** | 1 | The CL- related content makes me worried. | 1.69 | 1.281 | .876 | 1 | 1 |
| | 2 | Images of the CL cause me anxiety and confusion (tension, anxiety). | 1.71 | 1.296 | .876 | .97 | 1 |
| | 3 | The content of the CL frightens me. | 1.53 | 1.251 | .877 | .94 | 1 |
| | 4 | The content of the CL makes me nervous (angry and nervousness). | 1.32 | 1.265 | .876 | .92 | .57 |
| **8) Intention** | 1 | I intend to follow up on the prevention of CL. | 3.30 | 1.310 | .873 | .87 | .85 |
| | 2 | I intend to pay more attention to the educational content related to the CL. | 3.19 | 1.422 | .874 | 1 | 1 |
| | 3 | I plan to give my family the necessary suggestions on preventing CL. | 3.34 | 1.298 | .873 | .97 | 1 |
| | 4 | I intend to follow the preventive behaviors of the CL. | 3.33 | 1.269 | .873 | .97 | 1 |

(*Continued*)

**Table 1.** (Continued)

| Component | Row | Item | Mean | Std. Deviation | Cronbach's Alpha if Item Deleted | I-CVI | CVR |
|---|---|---|---|---|---|---|---|
| **9) Perceived Susceptibility in case of Insufficient Personal Protection** | 1 | I may get CL if I do not use an insect repellent pen or ointment at the time of the bite. | 1.98 | .822 | .874 | .92 | 1 |
| | 2 | I may also get CL. | 1.98 | .938 | .878 | 1 | 1 |
| | 3 | If the net is not installed on the doors and windows, I may get a CL. | 2.15 | .788 | .874 | .97 | 1 |
| | 4 | If I do not use a mosquito net to sleep outdoors, I may get CL. | 2.28 | .772 | .875 | 1 | 1 |
| **10) Individual Enablers** | 1 | I have been taught how to use an insect repellent pen. | 1.59 | 1.084 | .875 | 1 | .85 |
| | 2 | I have enough money to buy insect repellents, such as pens or repellent ointment. | 1.89 | 1.048 | .877 | 1 | .85 |
| | 3 | I know where to get insect repellents, such as pens or repellent ointment. | 2.22 | .835 | .875 | 1 | .85 |
| | 4 | I have been trained in preventive measures against CL. | 1.40 | 1.015 | .875 | 1 | .85 |
| **11) Environmental Enablers** | 1 | In our area, construction debris is collected quickly. | 1.76 | 1.014 | .876 | .97 | 1 |
| | 2 | In our area, trash is always put outside the house during the collection hours. | 2.18 | .942 | .876 | .97 | .85 |
| | 3 | My family installs a fine mesh fabric net in front of the air conditioner vent. | 1.85 | 1.180 | .875 | 1 | .85 |
| | 4 | My family installs fine-textured fabric netting in front of doors and windows. | 2.10 | 1.082 | .875 | 1 | 1 |
| **12) Perceived Likelihood of Bites at Any Height** | 1 | If I sleep on the bed, the sandfly cannot bite. | 1.98 | .882 | .879 | .97 | .85 |
| | 2 | If we live in the upper classes, I do not worry about getting CL. | 1.95 | .904 | .880 | .97 | .71 |

- The fourth factor, "Defensive Avoidance of the Message", with the variance of 4.640, consisted of five items. It indicated how much a student blocks the path for receiving messages about CL and avoids receiving such messages.

- The fifth factor was "Nurture", including five items with the variance of 3.381. It measured the role of other people in strengthening and encouraging a person in performing CL prevention behaviors.

- The sixth factor, "Perceived Severity", with the variance of 3.107, consisted of seven items related to students' perception of the severity of the disease and its complications.

- The seventh factor, "Fear", with the variance of 2.636, comprised four items and measured anxiety, anger, and worrisome caused by fear of CL messages and their complications.

- "Intention" was the eighth factor with the variance of 2.576. It included four items that measured the intention to perform preventive behaviors of CL.

- The ninth factor, "Perceived Susceptibility in case of Insufficient Personal Protection" with the variance of 2.325, included four items that measured the students' perception of the risk of CL in case of insufficient personal protection.

- The 10th factor was "Individual Enablers" with the variance of 2.123. It had four items that measured the students' access to facilities and their personal skills to prevent CL.

- The 11th factor, "Environmental Enablers", with the variance of 1.862, contained four items related to environmental facilities whose availability enabled the individual to prevent CL.

These include installing an efficient net into the doors, windows, and air conditioner vents, as well as a timely collection of construction waste and garbage by the related organizations.

- The 12th and last factor was "Perception of the Possibility of Bites at Any Height" with the variance of 1.787. It comprised two items about the students' knowledge about the fact that living on the upper floors or resting on the bed will also put them at risk of sandfly bites.

Factor analysis was performed to explore the main factors of the questionnaire. The factor load of each question as well as internal and external reliability factors, along with the Eigenvalue, Variancepercentage, Variance Cumulativepercentage are reported in Table 2.

### 3.3. Results of internal and external reliability assessment

The internal reliability of the instrument was calculated after factor analysis through Cronbach's alpha for the whole questionnaire and for each factor. The alpha coefficient for the 12 above-mentioned factors were 0.919, 0.888, 0.898, 0.852, 0.848, 0.813, 0.903, 0.913, 0.710, 0.653, 0.694, and 0.646, respectively; meanwhile, it was 0.877 for the whole instrument. Therefore, the internal reliability of the instrument was confirmed (Table 2).

The external reliability of the instrument was determined via ICC coefficient, which were 0.792, 0.767, 0.764, 0.712, 0.729, 0.740, 0.741, 0.710, 0.663, 0.817, 0.564, and 0.553 for the first to 12th factor, respectively. It was calculated as 0.833 for the whole instrument; thus, the external reliability of the instrument was confirmed (Table2).

## 4. Discussion

The present study aimed to develop and psychometrically assess an instrument for measuring CL prevention behavior and its related factors in adolescent females in endemic areas. The first phase of the study resulted in the development of a primary instrument consisting of 110 items, out of which 71 remained after the psychometric and exploratory factor analysis (EFA). In the majority of relevant studies in the research literature, 10 experts and fewer subjects participated in the analysis of the content validity of the instrument [13, 14, 26], but considering the complexity of leishmaniasis aspects, interdisciplinary participation of a higher number of experts seemed necessary for the present study; thus, 14 experts had invaluable contributions to the study. Concerning the reliability of the instrument, 0.7 was calculated as the acceptable cut-off in determining the external and internal reliability [25]. Accordingly, it can be said that the instrument has high reliability.

The previously proposed instruments mainly focused on individual factors and neglected social factors affecting CL prevention behaviors [9, 22]. They also skipped review of cultural and social factors while paying attention to the cultural context of behaviors seems critical, particularly in local outbreaks [24]. Even the tools did not pay enough attention to educational messages and fear reactions in CL-related [9, 14, 22], and these factors have been taken into account in the new instrument. In the construct validity assessment, 12 factors were extracted in the EFA stage, which will be discussed based on the PEN-3 Model and EPPM.

### 4.1. Extracted components based on EPPM

**4.1.1. Perceived efficacy.** "Perceived Efficacy" with 15 items was one of the extracted factors determined by EFA and based on the integration of "self-efficacy" and "response efficacy" items. The result of the integration of these two dimensions is called "Perceived Efficiency" in the EPPM [15, 16, 28]. The majority of previous CL-related instruments solely measured "Perceived Self-efficacy" [9, 22] and neglected "Perceived Response Efficacy". Although in other

**Table 2. Factor analysis results for the exploration of the main factors of the CL-preventive behaviors questionnaire and factors related to the integration of PEN-3 Model and EPPM, Cronbach's α coefficient, and ICC.**

| Abbreviation | Item | Perceived efficacy | Behavior | Message Minimization and Perceived Manipulation | Defensive Avoidance of the message | Nurtures | Perceived severity | Fear | Intention | Perceived susceptibility in insufficient personal protection | Individual enablers | Environmental enablers | The preception of the possibility of bites at any height |
|---|---|---|---|---|---|---|---|---|---|---|---|---|---|
| Self_E6 | 1 | .743 | | | | | | | | | | | |
| Self_E5 | 2 | .741 | | | | | | | | | | | |
| Self_E7 | 3 | .702 | | | | | | | | | | | |
| Response_5 | 4 | .695 | | | | | | | | | | | |
| Self_E2 | 5 | .670 | | | | | | | | | | | |
| Self_E3 | 6 | .667 | | | | | | | | | | | |
| Self_E9 | 7 | .667 | | | | | | | | | | | |
| Self_E8 | 8 | .661 | | | | | | | | | | | |
| Response_6 | 9 | .640 | | | | | | | | | | | |
| Response_3 | 10 | .639 | | | | | | | | | | | |
| Response_4 | 11 | .629 | | | | | | | | | | | |
| Self_E1 | 12 | .583 | | | | | | | | | | | |
| Response-2 | 13 | .578 | | | | | | | | | | | |
| Response-7 | 14 | .550 | | | | | | | | | | | |
| Self_E4 | 15 | .520 | | | | | | | | | | | |
| Behav7 | 16 | | .776 | | | | | | | | | | |
| Behave9 | 17 | | .700 | | | | | | | | | | |
| Behave3 | 18 | | .677 | | | | | | | | | | |
| Behave8 | 19 | | .673 | | | | | | | | | | |
| Behave6 | 20 | | .666 | | | | | | | | | | |
| Behave5 | 21 | | .654 | | | | | | | | | | |
| Behave4 | 22 | | .622 | | | | | | | | | | |
| Behave2 | 23 | | .552 | | | | | | | | | | |
| Behave1 | 24 | | .455 | | | | | | | | | | |
| FC_PM1 | 25 | | | .820 | | | | | | | | | |
| FC_PM2 | 26 | | | .797 | | | | | | | | | |
| FC_MM2 | 27 | | | .796 | | | | | | | | | |
| FC_MM3 | 28 | | | .765 | | | | | | | | | |
| FC_PM4 | 29 | | | .734 | | | | | | | | | |
| FC_PM3 | 30 | | | .687 | | | | | | | | | |
| FC_MM1 | 31 | | | .527 | | | | | | | | | |
| FC_DA6 | 32 | | | .503 | .471 | | | | | | | | |
| FC_DA2 | 33 | | | | .824 | | | | | | | | |
| FC_DA1 | 34 | | | | .761 | | | | | | | | |

Rotated Component Matrixa — Component

(*Continued*)

Table 2. (Continued)

**Rotated Component Matrixa**

| Abbreviation | Item | Perceived efficacy | Behavior | Message Minimization and Perceived Manipulation | Defensive Avoidance of the message | Nurtures | Perceived severity | Fear | Intention | Perceived susceptibility in insufficient personal protection | Individual enablers | Environmental enablers | The preception of the possibility of bites at any height |
|---|---|---|---|---|---|---|---|---|---|---|---|---|---|
| | | | | | | | | | | | | | Component |
| FC_DA4 | 35 | | | | .722 | | | | | | | | |
| FC_DA3 | 36 | | | | .708 | | | | | | | | |
| FC_DA5 | 37 | | | | .551 | | | | | | | | |
| Nurtures3 | 38 | | | | | .832 | | | | | | | |
| Nurtures2 | 39 | | | | | .810 | | | | | | | |
| Nurtures1 | 40 | | | | | .779 | | | | | | | |
| Nurtures5 | 41 | | | | | .704 | | | | | | | |
| Nurtures4 | 42 | | | | | .476 | | | | | | | |
| severity6 | 43 | | | | | | .695 | | | | | | |
| severity5 | 44 | | | | | | .662 | | | | | | |
| severity8 | 45 | | | | | | .640 | | | | | | |
| Severity7 | 46 | | | | | | .636 | | | | | | |
| Severity3 | 47 | | | | | | .599 | | | | | | |
| severity2 | 48 | | | | | | .576 | | | | | | |
| severity4 | 49 | | | | | | .534 | | | | | | |
| Fear3 | 50 | | | | | | | .889 | | | | | |
| Fear2 | 51 | | | | | | | .876 | | | | | |
| Fear1 | 52 | | | | | | | .868 | | | | | |
| Fear4 | 53 | | | | | | | .770 | | | | | |
| Intention3 | 54 | | | | | | | | .836 | | | | |
| Intention2 | 55 | | | | | | | | .816 | | | | |
| Intention4 | 56 | | | | | | | | .796 | | | | |
| Intention1 | 57 | | | | | | | | .788 | | | | |
| Susceptibil4 | 58 | | | | | | | | | .632 | | | |
| Susceptibil1 | 59 | | | | | | | | | .612 | | | |
| Susceptibil3 | 60 | | | | | | | | | .609 | | | |
| Susceptibil2 | 61 | | | | | | | | | .593 | | | |
| Enablers4 | 62 | | | | | | | | | | .644 | | |
| Enablers3 | 63 | | | | | | | | | | .627 | | |
| Enablers2 | 64 | | | | | | | | | | .508 | | |
| Enablers1 | 65 | | | | | | | | | | .504 | | |
| Enablers8 | 66 | | | | | | | | | | | .753 | |
| Enablers9 | 67 | | | | | | | | | | | .592 | |
| Enablers7 | 68 | | | | | | | | | | | .579 | |
| Enablers6 | 69 | | | | | | | | | | | .422 | |

(Continued)

Table 2. (Continued)

Rotated Component Matrix[a]

| | | Component | | | | | | | | | | | |
| Abbreviation | Item | Perceived efficacy | Behavior | Message Minimization and Perceived Manipulation | Defensive Avoidance of the message | Nurtures | Perceived severity | Fear | Intention | Perceived susceptibility in insufficient personal protection | Individual enablers | Environmental enablers | The preception of the possibility of bites at any height |
|---|---|---|---|---|---|---|---|---|---|---|---|---|---|
| susceptibil6 | 70 | | | | | | | | | | | | .810 |
| susceptibil7 | 71 | | | | | | | | | | | | .795 |
| Eigenvalue | | 14.874 | 6.466 | 4.265 | 3.294 | 2.400 | 2.206 | 1.872 | 1.829 | 1.651 | 1.507 | 1.322 | 1.269 |
| Variance% | | 20.949 | 9.107 | 6.006 | 4.640 | 3.381 | 3.107 | 2.636 | 2.576 | 2.325 | 2.123 | 1.862 | 1.787 |
| Variance Cumulative % | | 11.038 | 18.204 | 25.314 | 30.286 | 35.201 | 39.991 | 44.537 | 49.057 | 52.357 | 55.504 | 58.198 | 60.501 |
| ICC (%95CI) | | .792 (.612-.894) | .767 (.570-.880) | .764 (.566-.879) | .712 (.483-.850) | .729 (.510-.860) | .740 (.527-.866) | .741 (.529-.866) | .710 (.479-.849) | .663 (.408-.822) | .817 (.655-.908) | .564 (.267-.763) | .553 (.253-.757) |
| Cronbach's α | | .919 | .888 | .898 | .852 | .848 | .813 | .903 | .913 | .710 | .653 | .694 | .646 |

Extraction Method: Principal Component Analysis.

Rotation Method: Varimax with Kaiser Normalization.

[a]. Rotation converged in 9 iterations.

studies, it has been mentioned that some individuals believe in using a mosquito net while sleeping, installing a net on doors and window frames, or wearing long gowns to ward off sand flies as useful preventive measures [12, 21, 29, 30], no instrument has been developed for their assessment. "Perceived Efficacy" was measured for the first time in this questionnaire in order to determine the perception of individuals about the effectiveness of preventive behaviors and their confidence in the ability to put them in action. According to EPPM, positive perceptions regarding this aspect improves the likelihood of self-protective behavior [16]. Therefore, measuring this component using an appropriate instrument can play a major role in recognizing these perceptions, developing relevant appropriate interventions, and assessing the impact of interventions.

**4.1.2. Perceived severity.** "Perceived Severity" was another extracted component with seven items, expressing students' understanding of the severity of the disease and its complications. According to previous studies, this factor is one of the predictors of adopting preventive behaviors against CL, which has already been proposed based on the Health Belief Model [22]. However, according to the EPPM, this factor is among the predictors that indicate how people react to educational messages [15]. There are reports in the research literature implying that people sometimes consider CL lesions ugly and unpleasant; they regard the disease as a serious one with negative socioeconomic and aesthetic consequences, which reduces chance of getting married [3, 10, 31]. These notions may lead to maladaptive responses instead of preventive behaviors. Therefore, having such an instrument in the available armaments to measure this variable can be conducive to prevention of unpleasant consequences.

**4.1.3. Perceived susceptibility.** Perceived Susceptibility is the person's belief in vulnerability to the threat [15]. Perceived Susceptibility items were loaded in two factors. Four items measuring the students' perceptions of the risk of CL in the event of inadequate personal protection were labeled as "Perceived Sensitivity in case of Inadequate Personal Protection." The other two items were included in another factor called "Perception of the Possibility of Bites at Any Height" and indicated the false confidence in the safety against sandfly bites on upper floors or using the bed. Such misunderstanding causes people to be indifferent to preventive measures, such as installing a net on window frames or repellent devices and mosquito nets when resting on the bed. In this regard, a review study has shown only a few people in endemic areas believing that leishmaniasis is one of the health problems in their area and considering themselves at a high risk of this disease. Nevertheless, many believed that leishmaniasis was insignificant, which has led to negligence in taking preventive measures [21]. Although this factor is important, an extensive search revealed that the questionnaires with perceived susceptibility component have not paid attention to all the points [9]; meanwhile, it is of great necessity to provide appropriate tools to consider and assess this aspect in educational interventions.

**4.1.4. Danger control process.** According to the EPPM, if a person believes that the challenge is significant enough (perceived threat is high) and he/she believes in the efficiency of the behavior, they would deal with the problem and act upon it. This reaction is called "Danger Control Process", which controls the risk by changing the attitude and intention and finally changing the behavior [15, 16]. In the present questionnaire, two extracted factors, namely intention and behavior, were related to the risk control process, which will be discussed in next sections.

*4.1.4.1. Intention.* This component used four items to measure the individuals' intention to practice CL prevention behaviors. In a study by Ghodsi et al., it was reported that the intention, as a variable, is an indicator of a person's readiness to perform CL-preventive behaviors and is considered as an immediate predictor of the behavior [13]. Relevant searches resulted in the identification of "the intention to behavior" instrument which was based on the BASNEF

model [9, 14]. In our new instrument, this measurement could also indicate the success or failure of using EPPM [17]. If people do not intend to perform a behavior, they do not enter the danger control stage [23], which leads to the "no response" mode or even the fear control process. It should be noted that a validated questionnaire is of great necessity to determine whether the intention of the behavior contributes to practicing that behavior or preventing certain influential factors, such as defects in the enabling factors, which could impair the application of the behavior. Providing comprehensive questionnaires would make individuals capable of measuring all these dimensions, which was aimed and attempted to realize herein.

*4.1.4.2. Behavior.* This factor measures CL-preventive behaviors using nine items. Studies related to CL-preventive behaviors have shown the need to measure CL prevention behaviors [5, 13, 14]. Nevertheless, standard tools are needed to measure it. Behavior was measured based on the danger control process for the first time in the present study, where according to the threat and efficacy messages delivered to the person, how much a person seeks to observe CL prevention behaviors was assessed. Attention to these dimensions is critical since according to EPPM model, behaviors are determined based on their perceived efficiency against the threat [15, 32]; once the efficacy and threat are high in messages, individuals enter a danger control state and in the case of message efficacy, threat inconsistency leads to the fear and people will enter the fear control state [15]. In such a situation, they not only refrain from practicing preventive measures, but may also block the receiving of preventive advice and educational messages with the triple reactions to control the fear [16, 23].

**4.1.5. Fear.** "Fear" was one of the discovered factors with four items. The questionnaire in the present study is the first one to assess this aspect in the field of CL disease. Measuring this variable seems necessary since fear has, on a number of occasions, useful outcomes and it is regarded as an important motivator for preventive behavior although it is sometimes associated with failure [15]. Such failures are probably connected to the inability to integrate fear and practical recommendations based on Perceived Efficacy [33]. Therefore, evaluating these processes with new instrument contributes to the formulation and balancing the components of the message to prevent inconsistent and destructive reactions [23].

**4.1.6. Fear control process.** As mentioned earlier and according to the EPPM, behaviors are determined by perceived effectiveness against the threat [32]. If a person does not believe in the effectiveness of preventive behaviors and considers himself incapable of overcoming the problem, he/she escapes the problem and rejects the health-associated messages; this reaction is called the fear control process [15]. The three different responses in the fear control domain are defensive avoidance of the message, message minimization, and perceived manipulation [28]. Thus, the next two extracted factors, which will be discussed, are related to the fear control process.

*4.1.6.1 Message minimization and perceived manipulation.* Message Minimization and Perceived Manipulation, with eight items, measured two types of maladaptive reactions to educational messages. The first one was massage minimization by devaluing and discrediting the massage; the second reaction was perceived as manipulation [23]. For the first time, in a CL-related tool, the items measuring these variables were adapted to receiving messages to measure the individuals entering the fear control process and non-compliance with preventive behavior. Due to the created fear, people may devalue the messages and warning content and consider them as false and distorted messages [17].

*4.1.6.2 Defensive avoidance of the message.* This extracted component was also a different type of individuals' reaction to fear control. These reactions are a kind of maladaptive ones against risk massages or messengers [23, 34]. To date, no instrument has been developed to measure this variable regarding the determination of the approach of the society to CL prevention. Having the instrument at hand, when people feel fear and perceive that the behavior is

not effective, or once they are not able to observe the behavior and refrain from receiving the risk prevention message by any means, it is possible to proceed for correct educational messages and interventions while evaluating their results. The present instrument measures students' avoidance of CL-related messages and their risks; this assessment helps the implementation of more effective interventions in the future so that fewer people experience "fear control" reactions.

## 4.2. Extracted components based on PEN-3 model

The dimensions of the PEN-3 model in this questionnaire included Perceptions, Enablers, and Nurture. The Perceptions were consistent with Perceived Efficacy, Perceived Susceptibility, and Perceived Severity in the EPPM, as described in the previous sections.

**4.2.1. Nurture.** This component consisted of five items that measured the support and encouragement by others in performing CL prevention behaviors. Encouragement and support from others are among behavior-promoting factors. Individual judgments of the risk and the importance of observing preventive behaviors are also influenced by the encouragement and the behaviors of the others, which could be evaluated with a standard questionnaire. To the best of our knowledge, there were no tools for measuring the promoters for our subject. However, in certain studies [13, 14], some items were identified, which were adjusted for subjective norms and similar to the nurtures item in the present study. Subjective norms emerges as a social pressure by others (friends, relatives, family members, and health care systems) on the individual to perform a particular behavior [35]. It means that the presence of incentive people is a factor in practicing preventive behaviors and conversely, lack of support from influential people can promote fear control reactions. The importance of subjective norms in performing CL preventive behaviors has been also highlighted in other studies [13, 14]. Thus, by measuring this factor, we will identify those people around students who are prone to intervention, in addition to the students themselves.

**4.2.2. Individual enablers and environmental enablers.** Assessment of Enabler items was based on two components. Four items measuring the access to facilities and personal skills to prevent CL were labeled as "individual enablers". Moreover, four items related to the environmental facilities, which helped the individual in CL prevention, were included in another component called "environmental enablers". In the relevant research literature, Enablers are referred to as factors influencing and predicting protective behaviors, including preventive behaviors against CL [13, 18]. Searching the published resources in this regard, we found measuring tools of Enablers based on the BASNEF model [9, 14, 18]. In the study of Ghodsi et al. in field of psychometric of assessment tool of students' preventive behaviors, five questions were posed as follows: 1- Existence of financial resources to purchase mesh net, insect repellent, insecticides; 2- Access to information on prevention behaviors of leishmaniasis; 3- Access to doctors in health centers; 4- Access to dermatologists; 5—Access to health personnel to learn methods of leishmaniasis prevention [9]. In the present study, we tried to design only preventive items. After performing factor analysis, they were classified into two areas: individual and environmental. The items were developed based on a qualitative study consistent with the PEN-3 Model and other qualitative studies, which indicated that people could not sometimes observe preventive behaviors due to certain obstacles, such as the lack of skills and resources [18, 21, 30]. This means that the availability of skills and resources facilitates the preventive behavior practice while lack of resources and lack of access to protective facilities would trigger fear control reactions. As a result, it is necessary to measure this factor with a valid tool according to the context.

### 4.3. Study limitations

One of the difficulties faced conducting the present study was the COVID-19 pandemic that prevented face-to-face interviews for face validity; it was performed via phone calls. Furthermore, we had to complete the questionnaires online instead of doing so in person. Failure to perform confirmatory factor analysis was another limitation herein. It could be suggested that confirmatory factor analysis be performed in future studies.

## 5. Conclusion

The present questionnaire integrated a Cultural Model and an individual message design model, which are the first steps for assessment and measurement of the impact of educational interventions, especially the impact of educational messages on CL prevention behaviors and the related factors. By applying this tool in other cultures and its implementation, educational priorities based on individual and social factors could be set to change the preventive behaviors of CL after assessing the needs in endemic areas. Subsequently, effective training would be designed, implemented, and evaluated considering the educational needs of each region in order to eliminate the negative factors and strengthen the positive ones. This tool could help researchers and healthcare professionals to assess the type of audience's favorable or unfavorable reaction to educational messages. Therefore, based on the results of this assessment, they will be able to formulate the elements of educational messages so that they are as effective as possible.

## Supporting information

**S1 Dataset.**
(SAV)

## Acknowledgments

The authors appreciate the participation of the students and express their gratitude for their trust in researchers.

## Author Contributions

**Conceptualization:** Hossein Shahnazi, Fereshteh Zamani-Alavijeh.

**Data curation:** Masoumeh Alidosti.

**Formal analysis:** Zahra Heidari.

**Funding acquisition:** Fereshteh Zamani-Alavijeh.

**Investigation:** Masoumeh Alidosti.

**Methodology:** Hossein Shahnazi, Fereshteh Zamani-Alavijeh.

**Project administration:** Fereshteh Zamani-Alavijeh.

**Resources:** Fereshteh Zamani-Alavijeh.

**Software:** Zahra Heidari.

**Validation:** Hossein Shahnazi, Zahra Heidari.

**Visualization:** Hossein Shahnazi, Zahra Heidari.

**Writing – original draft:** Masoumeh Alidosti, Fereshteh Zamani-Alavijeh.

**Writing – review & editing:** Masoumeh Alidosti, Hossein Shahnazi, Zahra Heidari, Fereshteh Zamani-Alavijeh.

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
