## [Decision Letter · Decision Letter 0]

12 Jan 2022

PONE-D-21-33175Development and Psychometric Assessment of Cutaneous Leishmaniasis Prevention Behaviors questionnaire in adolescent female students: Application of Integration of Cultural Model and Extended Parallel Process Model.PLOS ONE

Dear  Dr. Alavijeh,

Thank you for submitting your manuscript to PLOS ONE. After careful consideration, we feel that it has merit but does not fully meet PLOS ONE’s publication criteria as it currently stands. Therefore, we invite you to submit a revised version of the manuscript that addresses the points raised during the review process.

We look forward to receiving your revised manuscript.

Kind regards,

Mumtaz Alam, PhD

Academic Editor

PLOS ONE

Journal Requirements:

“NO-Include this sentence at the end of your statement: The funders had no role in study design, data collection and analysis, decision to publish, or preparation of the manuscript.”

Reviewers' comments:

Reviewer's Responses to Questions

**Comments to the Author**

1. Is the manuscript technically sound, and do the data support the conclusions?

Reviewer #1: Yes

Reviewer #2: Yes

2. Has the statistical analysis been performed appropriately and rigorously? 

Reviewer #1: Yes

Reviewer #2: Yes

3. Have the authors made all data underlying the findings in their manuscript fully available?

Reviewer #1: Yes

Reviewer #2: Yes

4. Is the manuscript presented in an intelligible fashion and written in standard English?

Reviewer #1: Yes

Reviewer #2: No

5. Review Comments to the Author

Reviewer #1: I read the manuscript about “Development and Psychometric Assessment of Cutaneous Leishmaniasis Prevention Behaviors questionnaire in adolescent female students: Application of Integration of Cultural Model and Extended Parallel Process Model.” This research is a fascinating study about one of the essential points facing leishmaniasis: prevention behavior. One of the positive points of this study was the integration of two cultural and individual models n, which according to the social structure of the study area, can be effective in investigating prevention behavior. A suitable questionnaire has been designed using the opinions of health education specialists and healthcare workers together as an expert in this field. Considering the location of Isfahan as one of the most critical endemics of leishmaniasis in Iran, a careful study of residents' opinions regarding prevention behaviors can be of great help to health system planners

The authors developed a preliminary questionnaire based on the integration of cultural PEN-3 and Extended parallel models and were completed by 460 adolescent female students resident in endemic areas of zoonotic CL IN Isfahan. However, minor revisions are needed before being considered for publication in PloS One.

- Mixed capital and small letters are used in the title. The title should be corrected based on the journal's style.

- Mixed capital and small letters have also been inconsistently used in the subtitles (parts 3.1, 3.2, 3.3, and others).

- Some terms should be definied at first appearance (i.e., PEN, ICC should be defined at previous parts).

- A thorough revision of the text is critical.

- L69, currently, the official incidence rate is 10,000 to 15000/100,000 persons annually.

- The authors should elaborate on the study area, the causative species, and the disease's burden.

- The prepared diagrams is not sharp

Reviewer #2: Dear Editor:

Thank you for concerning me to review the manuscript, entitled “Development and Psychometric Assessment of Cutaneous Leishmaniasis Prevention Behaviors questionnaire in adolescent female students: Application of Integration of Cultural Model and Extended Parallel Process Model.”

The aim of this article is to development and psychometric assessment of Cutaneous Leishmaniasis prevention behaviors questionnaire with application of integration of cultural PEN-3 Model and Extended Parallel Process Model in adolescent female students in endemic areas. Overall, the study is interesting and novel; however, the following concerns need to be corrected:

Title:

- According to the format of journal, all the first letters of the words except the first word should be written in lower case. “Development and psychometric assessment of cutaneous leishmaniasis prevention behaviors questionnaire in adolescent female students: application of integration of cultural model and extended parallel process model”

Abstract:

- Please explain more about the aim of the study in the abstract section.

- Line 48; write the word “Test-retest” in lower case.

Introduction:

- The manuscript needs extensive stylistic and English editing by a native speaker with science background.

- Line 69; use “incidence” instead of “incident” and in recent references; Iran is one of the seven countries with the highest prevalence of cutaneous leishmaniasis (CL). Correct this sentence with new reference.

o Mehdi Bamorovat , Iraj Sharifi , Esmat Rashedi, Alireza Shafiian , Fatemeh Sharifi , Ahmad Khosravi , Amirhossein Tahmouresi . A novel diagnostic and prognostic approach for unresponsive patients with anthroponotic cutaneous leishmaniasis using artificial neural networks. PloS One. 2021;16(5): e0250904.

- Line 81: Correct “CL” with upper cases.

- Line 84: Use “CL” instead of “cutaneous leishmaniasis”.

- Line 104: Cite references after “…. environments to prevent CL”.

- Line 105: Figure 1 is not sharp. Increase the resolution of Figure 1.

Method:

- Mention the age range of participants.

- Figure 2 is not sharp at all. Increase the resolution of that.

- Line 150: were the students who requested to comment on the importance level of questions, infected to cutaneous leishmaniasis?

- Line 211: Correct “0.889”

Results:

- Table 1: the questions of the first factor "Perceived efficacy” are not enough and useful and they are just repeated and are not enough for effectiveness of the recommended behaviors.

- Line 219,224,239,229,231,233 and …: Correct the decimal point “/” to “.” Like 1.895.

- Table 1. In the ninth factor, "Perceived Susceptibility in case of Insufficient Personal Protection" Is question 2 proper for this section?

- Explain more about table 2.

Discussion

- Line 7: Write the “Leishmaniasis” with lower cases.

- In part 4-1-3 “Perceived susceptibility”: compared your results with another studies such as this review study “Behaviors and Perceptions Related to Cutaneous Leishmaniasis in Endemic Areas of the World: A Review”

- Part 4-2-2” Individual enablers and environmental enablers”: in this part you explained about results and questionnaire, please compared with another studies.

Conclusion

- Explain more about the message you want to deliver.

Availability of data and materials

- Line 162: correct the “fles” to “files”

6. PLOS authors have the option to publish the peer review history of their article (what does this mean?). If published, this will include your full peer review and any attached files.

Reviewer #1: **Yes: **Ehsan Salarkia

Reviewer #2: No

---

## [Author Response · Author response to Decision Letter 0]

1 Feb 2022

Date: 27 January 2022

From: "Fereshteh Zamani-Alavijeh" fe.zamani@gmail.com

To: 

"Journal PLoS ONE " 

Title: Development and psychometric assessment of cutaneous leishmaniasis prevention behaviors questionnaire in adolescent female students: application of integration of cultural model and extended parallel process model.

Dear Editor:

We appreciate the thoughtful comments of your reviewers, which have added much value to the current revised version. 

In this letter, we have listed a point-by-point response to each comment. We also have considered the journal style and instructions in the revised paper. All changes are marked in the new version.

1. The manuscript complies with PLOS ONE style requirements, including file naming.

2. The grant information you provided in the ‘Funding Information’ and ‘Financial Disclosure’ sections do match.

3- Isfahan University of Medical Sciences, IRAN. Supported our study but did not provide funding for this study.

4- The supporters had no role in study design, data collection and analysis, decision to publish, or preparation of the manuscript.

 5- The authors received no specific funding for this work.

6. Upon re-submitting revised manuscript, we upload the study underlying data set as either Supporting Information files.

Reviewer1:

--- Mixed capital and small letters are used in the title. The title should be corrected based on the journal's style.

Author' Response: Thank you for your comment. Was corrected.

--- Mixed capital and small letters have also been inconsistently used in the subtitles (parts 3.1, 3.2, 3.3, and others).

Author' Response: Thank you for your comment. Was corrected.

--- Some terms should be definied at first appearance (i.e., PEN, ICC should be defined at previous parts).

Author' Response: Thank you for your comment. 

-Intraclass correlation coefficient (ICC) index, was written in the abstract, the method part.

- PEN-3 term definied in introduction on page 4 lines 103 to 107. The PEN-3 model consists of three domains: (1) Cultural Identity, (2) Relationships and Expectations, and (3) Cultural Empowerment. Each domain includes three factors that form the acronym PEN; Person, Extended Family, Neighborhood (Cultural Identity domain); Perceptions, Enablers, and Nurturers (relationship and expectation domain); Positive, Existential and Negative (Cultural Empowerment domain).

--- A thorough revision of the text is critical.

Author' Response: Thank you for your comment. The text was revised again.

--- Currently, the official incidence rate is 10,000 to 15000/100,000 persons annually.

Author' Response: Thank you for your comment. According to the latest searches conducted on the World Health Organization website and the latest articles published in reputable journals: 

It is estimated that between 600 000 to 1 million new CL cases occur worldwide annually. (World Health Organization. Leishmaniasis—Key facts. 8 January 2022. https://www. who.int/en/news-room/fact-sheets/detail/leishmaniasis. Accessed 13 January 2022. https://www.who.int/news-room/fact-sheets/detail/leishmaniasis), and Mendizábal-Cabrera R, Pérez I, Becerril Montekio V, Pérez F, Durán E, Trueba ML. Cutaneous leishmaniasis control in Alta Verapaz (northern Guatemala): evaluating current efforts through stakeholders’ experiences. Infectious Diseases of Poverty [Internet]. 2021 Dec 7;10(1):61. Available from: https://doi.org/10.1186/s40249-021-00842-3

This was corrected in the introduction of line 73.

--- The authors should elaborate on the study area, the causative species, and the disease's burden.

Author' Response: Thank you for your comment. This was corrected in the introduction of line 76 to 79.Isfahan located in the center of Iran on green plains of Zayandeh Rud River, is one of the CL endemic provinces in Iran, which has faced a significant increase in the prevalence of the disease over the last decade. There is particularly in the north and northeast Isfahan province, the wet or rural type (zoonotic leishmaniasis or ZCL). ZCL caused by leishmania major and transmitted by Phlebotomus papatasi (7,8)

--- The prepared diagrams is not sharp

Author' Response: Thank you for your comment, Resolution increased of diagrams 1 and 2.

Reviewer #2:

Title:

--- According to the format of journal, all the first letters of the words except the first word should be written in lower case. 

Author' Response: Thank you for your comment, was corrected.

 “Development and psychometric assessment of cutaneous leishmaniasis prevention behaviors questionnaire in adolescent female students: application of integration of cultural model and extended parallel process model”

Abstract:

--- Please explain more about the aim of the study in the abstract section.

Author' Response: Thank you for your comment, was corrected.

Cutaneous Leishmaniasis (CL) is an important public health issue in least 83 countries, including Iran. Individuals’ behavior modification is believed to be one of the best ways for CL prevention. However, a comprehensive questionnaire for identification of CL prevention behaviors and its numerous associated factors, as well as the impact of educational messages, has not yet been psychoanalyzed. So the present study was conducted to development and psychometric assessment of cutaneous leishmaniasis prevention behaviors questionnaire in female students.

--- Line 48; write the word “Test-retest” in lower case.

Author' Response: Thank you for your comment, was corrected.

Introduction: 

--- The manuscript needs extensive stylistic and English editing by a native speaker with science background.

Author' Response: Thank you for your comment. The text was revised again.

--- Line 69; use “incidence” instead of “incident” and in recent references; Iran is one of the seven countries with the highest prevalence of cutaneous leishmaniasis (CL). Correct this sentence with new reference.

Mehdi Bamorovat , Iraj Sharifi , Esmat Rashedi, Alireza Shafiian , Fatemeh Sharifi , Ahmad Khosravi , Amirhossein Tahmouresi . A novel diagnostic and prognostic approach for unresponsive patients with anthroponotic cutaneous leishmaniasis using artificial neural networks. PloS One. 2021;16(5): e0250904.

Author' Response: Thank you for your comment. Was corrected.

With an annual incidence rate of 30,000 cases(5), Iran is one of the seven countries with the highest prevalence of CL(6).

--- Line 81: Correct “CL” with upper cases.

--- Line 84: Use “CL” instead of “cutaneous leishmaniasis”.

Author' Response: Thank you for your comment. Both were corrected

---Line 104: Cite references after “…. environments to prevent CL”.

Author' Response: Thank you for your comment, was corrected.

In addition to perceptions(perceived susceptibility, perceived severity, and self-efficacy,..) this cultural model considers incentives or “nurtures” and also “enablers”, for preventive behaviors (20)

--- Line 105: Figure 1 is not sharp. Increase the resolution of Figure 1.

Author' Response: Thank you for your comment, increased resolution of Figure 1 and 2.

Method: 

- Mention the age range of participants.

Author' Response: Thank you for your comment, was corrected. )were 12 to 17 years(

- Figure 2 is not sharp at all. Increase the resolution of that.

Author' Response: Thank you for your comment, Resolution increased of Figure 1 and 2.

- Line 150: were the students who requested to comment on the importance level of questions, infected to cutaneous leishmaniasis?

Author' Response: The 10 students, who commented on the importance of the level of questions, lived in endemic areas, themselves or a family member or friend or distant relative was infected.

- Line 211: Correct “0.889”

Author' Response: Thank you for your comment, was corrected.

Results:

--- Table 1: the questions of the first factor "Perceived efficacy” are not enough and useful and they are just repeated and are not enough for effectiveness of the recommended behaviors.

Author' Response: Thank you for your comment. Perceived efficiency includes response efficacy and self-efficacy. The result of the integration of these two dimensions is called “Perceived Efficiency” in the extended Parallel Process Model. The questions are written in order of factor load in the table and may seem repetitive at first, but measure two different dimensions of response efficacy and self-efficacy.

Response efficacy questions are: 

4- My request to the family for installing a suitable net for doors and windows and air conditioner vents will be effective.

9- My request from the family to repair cracks in the walls of the house will be effective.

10-The use of insect repellents, such as ointment and repellent pen, is useful to prevent CL.

11- My request to my family to provide insect repellents, such as a repellent pen, will be effective.

13- The use of mosquito nets at rest is useful to prevent CL.

14- It would be effective for my family to contact the municipality or relevant organizations to collect construction debris.

These items measure a person's belief in the effectiveness of recommended behaviors, that an important mediator between self-efficacy and behavior.

Self-efficacy questions are:

1- I can ask my family to install a suitable net in door frames, window frames, and air conditioner vents.

2- I can ask my family to provide insect repellents, such as spray and repellent pen.

3- I can ask my family to repair the cracks on the walls of the house.

5- I can use an insect repellent pen or ointment properly.

6- I can use a mosquito net to sleep outdoors.

7- I can tell my family to put the garbage out of the house during the collection hour.

8- My family can contact the municipality or relevant organizations to collect construction waste.

12- I can cover most parts of the body during biting times.

15- I can avoid sleeping outdoors if I do not have a mosquito net.

These items measure a person's belief about in the ability to perform behavior.

--- Line 219,224,239,229,231,233 and …: Correct the decimal point “/” to “.” Like 1.895.

Author' Response: Thank you for your comment. Was corrected.

- Table 1. In the ninth factor, "Perceived Susceptibility in case of Insufficient Personal Protection" Is question 2 proper for this section?

Author' Response: Thank you for your comment. Perceived susceptibility is the person’s belief in vulnerability to the threat. likelihood of having a disease: "I may also get".

Shirahmadi S, Seyedzadeh-Sabounchi S, Khazaei S, Bashirian S, Miresmæili AF, Bayat Z, et al. Fear control and danger control amid COVID-19 dental crisis: Application of the Extended Parallel Process Model. Kielbassa AM, editor. PLOS ONE [Internet]. 2020 Aug 13;15(8):e0237490. Available from: http://dx.doi.org/10.1371/journal.pone.0237490.

In this study, susceptibility was measured with more detail in 6 questions. With factor analysis, the questions were divided into two categories and naming was as follows:; "Perceived Sensitivity in case of Inadequate Personal Protection" and "Perception of the Possibility of Bites at Any Height".

--- Explain more about table 2.

Author' Response: Thank you for your comment. Factor analysis was performed for the exploration of the main factors of the questionnaire. The factor load of each question as well as of internal and external reliability the factors, and the Eigenvalue, Variancepercentage, Variance Cumulativepercentage are reported in Table 2. 

Discussion

--- Line 7: Write the “Leishmaniasis” with lower cases.

Author' Response: Thank you for your comment. Was corrected.

--- In part 4-1-3 “Perceived susceptibility”: compared your results with another studies such as this review study “Behaviors and Perceptions Related to Cutaneous Leishmaniasis in Endemic Areas of the World: A Review

Author' Response: Thank you for your comment. Was corrected. In this regard, a review study has shown only a few people in endemic areas believe that leishmaniasis is one of the health problems in area and consider themselves a high risk group for this disease. While many believed that leishmaniasis was insignificant and this led to negligence in taking preventive measures.(21). Although this factor is Important, but an extensive search revealed that the questionnaires with perceived susceptibility component have not paid attention to all the points (9)

- Part 4-2-2” Individual enablers and environmental enablers”: in this part you explained about results and questionnaire, please compared with another studies.

Author' Response: Thank you for your comment. Was corrected.

In the study of Ghodsi et al, in field of psychometric of assessment tool of students' preventive behaviors, 5 questions were posed as follows: 1- Existence of financial resources to purchase mesh net , insect repellent, insecticides, 2- Access to information on prevention behaviors of leishmaniasis, 3- Access to doctors in health centers, 4- Access to Dermatologist, 5 - Access to health personnel to learn methods to prevention leishmaniasis (9). In the present study tried to be designed only preventie items. That after performing factor analysis, it was classified into two areas: individual and environmental.

Conclusion

--- Explain more about the message you want to deliver.

Author' Response: Thank you for your comment. Was corrected.

The present questionnaire integrates a cultural model and an individual message design model and the first step is for assessment and measurement of the impact of educational interventions, especially the impact of educational messages on CL prevention behaviors and the related factors. By applying this tool in other cultures and its implementation, educational priorities based on individual and social factors could be set to change the preventive behaviors of CL after assessing the needs in endemic areas. Subsequently, effective training would be designed, implemented, and evaluated considering the educational needs of each region in order to eliminate the negative factors and strengthen the positive ones. This tool helps researchers and healthcare professionals to assess the type of audience's favorable or unfavorable reaction to educational messages. Therefore, based on the results of this assessment, they will be able to formulate the elements of educational messages to be as effective as possible.

---Availability of data and materials

Author' Response: With the permission of the ethics Committee of Isfahan University of Medical Sciences, was upload the study underlying data set as either Supporting Information files.

--- Line 162: correct the “fles” to “files”

Author' Response: Thank you for your comment. Was corrected.

---

## [Decision Letter · Decision Letter 1]

9 Aug 2022

Development and psychometric assessment of cutaneous leishmaniasis prevention behaviors questionnaire in adolescent female students:  application of integration of cultural model and extended parallel process model

PONE-D-21-33175R1

Dear Dr. Zamani-Alavijeh,

We’re pleased to inform you that your manuscript has been judged scientifically suitable for publication and will be formally accepted for publication once it meets all outstanding technical requirements.

Kind regards,

Mona Dür, PhD, MSc

Academic Editor

PLOS ONE

Reviewers' comments:

Reviewer's Responses to Questions

**Comments to the Author**

1. If the authors have adequately addressed your comments raised in a previous round of review and you feel that this manuscript is now acceptable for publication, you may indicate that here to bypass the “Comments to the Author” section, enter your conflict of interest statement in the “Confidential to Editor” section, and submit your "Accept" recommendation.

Reviewer #1: All comments have been addressed

Reviewer #2: All comments have been addressed

2. Is the manuscript technically sound, and do the data support the conclusions?

Reviewer #1: Yes

Reviewer #2: Yes

3. Has the statistical analysis been performed appropriately and rigorously? 

Reviewer #1: Yes

Reviewer #2: I Don't Know

4. Have the authors made all data underlying the findings in their manuscript fully available?

Reviewer #1: Yes

Reviewer #2: Yes

5. Is the manuscript presented in an intelligible fashion and written in standard English?

Reviewer #1: Yes

Reviewer #2: Yes

6. Review Comments to the Author

Reviewer #1: I read the revised manuscript about “Development and Psychometric Assessment of Cutaneous Leishmaniasis Prevention Behaviors questionnaire in adolescent female students: Application of Integration of Cultural Model and Extended Parallel Process Model.”

All comments made by the author have been applied and in my opinion are approved for publication in Plus One journal.

Reviewer #2: (No Response)

7. PLOS authors have the option to publish the peer review history of their article (what does this mean?). If published, this will include your full peer review and any attached files.

Reviewer #1: No

Reviewer #2: No

---

## [Editor Report · Acceptance letter]

11 Aug 2022

PONE-D-21-33175R1 

Development and psychometric assessment of cutaneous leishmaniasis prevention behaviors questionnaire in adolescent female students:  application of integration of cultural model and extended parallel process model 

Dear Dr. Zamani-Alavijeh:

I'm pleased to inform you that your manuscript has been deemed suitable for publication in PLOS ONE. Congratulations! Your manuscript is now with our production department. 

Kind regards, 

on behalf of

Dr. Mona Dür 

Academic Editor

PLOS ONE